# Multiple Brain Abscesses of Odontogenic Origin. May Oral Microbiota Affect Their Development? A Review of the Current Literature

**Nicola Montemurro** [1,2], **Paolo Perrini** [1,2], **Walter Marani** [3], **Bipin Chaurasia** [4], **Massimo Corsalini** [5], **Antonio Scarano** [6] **and Biagio Rapone** [7,*]

1. Department of Neurosurgery, Azienda Ospedaliera Universitaria Pisana (AOUP), 56100 Pisa, Italy; nicola.montemurro@unipi.it (N.M.); paolo.perrini@unipi.it (P.P.)
2. Department of Translational Research and of New Surgical and Medical Technologies, University of Pisa, 56126 Pisa, Italy
3. Unit of Neurosurgery, Mater Dei Hospital, 70121 Bari, Italy; wmarani1@gmail.com
4. Department of Neurosurgery, Bangladesh State Medical University, Dhaka 1205, Bangladesh; troexa@gmail.com
5. Interdisciplinary Department of Medicine, University of Bari, 70121 Bari, Italy; massimo.corsalini@uniba.it
6. Department of Oral Science, Nano and Biotechnology and CeSi-Met University of Chieti-Pescara, 66100 Chieti, Italy; ascarano@unich.it
7. Department of Basic Medical Sciences, Neurosciences and Sense Organs, "Aldo Moro" University of Bari, 70121 Bari, Italy
* Correspondence: biagio.rapone@uniba.it

**Abstract:** In the last few years, the role of oral microbiota in the setting of oral diseases such as caries, periodontal disease, oral cancer and systemic infections, including rheumatoid arthritis, cardiovascular disease and brain abscess (BA), has attracted the attention of physicians and researchers. Approximately 5–7% of all BAs have an odontogenic origin, representing an important pathological systemic condition with a high morbidity and mortality. A systematic search of two databases (Pubmed and Ovid EMBASE) was performed for studies published up to 5 January 2021, reporting multiple BAs attributed to an odontogenic origin. According to PRISMA guidelines, we included a total of 16 papers reporting multiple BAs due to odontogenic infections. The aim of this review is to investigate the treatment modality and the clinical outcome of patients with multiple BAs due to odontogenic infections, as well as to identify the most common pathogens involved in this pathological status and their role, in the oral microbiota, in the onset of oral infections. A multidisciplinary approach is essential in the management of multiple BAs. Further studies are required to understand better the role of microbiota in the development of multiple BAs.

**Keywords:** brain abscess; odontogenic; dental origin; central nervous system infection; dental infection; cerebral abscess; oral infection; neurosurgery; surgery

## 1. Introduction

Brain abscess (BA), defined as a focal infection within the brain parenchyma, which starts as a localized area of cerebritis and subsequently is converted into a collection of pus within a well-vascularized capsule [1], represents a universal health problem with a high long-term morbidity, and a mortality rate of up to 9.5% reported in some studies [2–4]. The reported incidence of BAs was estimated at 0.3 to 1.3 cases per 100,000 people per year [5], with a male predominance (ratio resulted from 2:1 to 3:1) and a median age of 30 to 40 years in most pediatric and adult series, although the age distribution varies depending on the predisposing condition leading to the formation of BA [5]. Approximately 5–7% of all BAs are caused by dental infection and manipulation. Dental procedures in the setting of periodontal disease can provoke focal oral infections. Multiple abscesses account for 5–50% of all BAs, which are often seen, but not only, in immunocompromised patients and

are the result of hematogenous spread. Some microorganisms are more common causative than others; a well-known condition, although not fully clarified [6,7].

The human oral cavity contains a very broad range of microorganisms. These microorganisms constitute the human oral microbiota, which represents one of the most complex microbial communities in the human body [8]. Some authors reported about 500 different species and 700 kinds of microorganisms [8], and that all known microorganisms associated with humans are at some time found in the oral cavity as either transient (the majority) or resident (only a few) species [9]. It seems that the bacteria that cause odontogenic infections are generally saprophytes. Some microorganisms are usually more responsible for these illnesses compared to others (in well-known conditions, still to be clarified) [10].

In recent years, the role of oral microbiota in the setting of oral diseases such as caries, periodontal disease, oral cancer and systemic infections has attracted the attention of physicians and researchers [11–15]. There is also evidence that oral microbiota is closely related to some systemic diseases, including rheumatoid arthritis, adverse pregnancy outcomes, cardiovascular disease and BA [16–21]. In addition, a recent study reported the role of saliva and saliva early mediators of oral microbiota, like serum and salivary Galectin-3 levels, in predicting periodontitis [21–23]. Patients suffering from severe untreated periodontal disease frequently experience bacteraemia after tooth brushing, flossing and chewing [23]. Frequent bacteraemia and systemic spill of proinflammatory cytokines [24] from periodontal pockets result in the release of leukocyte elastase and acute phase proteins. The link between oral infections and adverse systemic conditions such as multiple BAs has attracted much attention in the research community in the last few years and with it, also the role of oral microbiota and the mechanism underlying its spreading from mouth to brain, through transient bacteremia resulting in bacterial colonization in extra-oral sites. It seems that the composition of primary odontogenic infections reflects the normal oral microbiota and the method of dissemination.

To the best of our knowledge, this is the first review of multiple BAs of odontogenic origin. The aim of this review is to assess the more common pathogens involved in multiple BAs, the possible surgical and medical treatments, the antibiotic therapies taken into account and the role of microbiota in the onset of oral infections and blood dissemination, leading to the development of multiple BAs.

## 2. Materials and Methods

### 2.1. Literature Search

A Pubmed and Ovid EMBASE search was performed to identify articles from the period 1988 to present, relevant to multiple brain abscesses attributed to an odontogenic origin. PRISMA guidelines (Preferred Reporting Items for Systematic Reviews and Meta-analyses) were followed [25]. The key words "multiple brain abscess", "multiple cerebral abscess", "odontogenic", "dental" and "dental origin" were used in both "AND" and "OR" combinations. The key words and the detailed search strategy are reported in Table 1.

**Table 1.** Search syntax.

| PubMed Search Accessed on 5 January 2021 (42 Articles) | Embase Search Accessed on 5 January 2021 (46 Articles) |
| --- | --- |
| (multiple) AND (brain abscess OR brain abscesses OR cerebral abscess OR cerebral abscesses) AND (odontogenic OR dental OR dental origin) | ('multiple') AND ('brain abscess' OR 'brain abscesses' OR 'cerebral abscess' OR 'cerebral abscesses') AND ('odontogenic' OR 'dental' OR 'dental origin') |

The inclusion criteria were the following: case series or case reports reporting neurological clinical data, pathogens and outcome of patients with multiple BAs due to odontogenic infections. Exclusion criteria were the following: (1) studies published in languages other than English with no available English translations, (2) review articles, (3) case series or

case report reporting single BA, (4) studies that did not involve human beings, (5) studies with insufficient data, (6) studies not related with this topic.

### 2.2. Data Collection

From each study, we extracted the following: (1) patient's demographics; (2) neurological clinical presentation; (3) number of BAs and their locations; (4) pathogens involved; (5) treatment modality; (6) clinical outcome.

### 2.3. Outcomes

The primary objective of this systematic review was to examine the treatment modality and the clinical outcome of patients with multiple BAs due to odontogenic infections. The secondary objectives were to identify the most common clinical presentation and the most common pathogens involved in this pathological status.

## 3. Results

### 3.1. Literature Review

The database search yielded 88 articles. After the removal of duplicates, 47 articles were eligible for screening. A total of 16 articles met the selection criteria [6,26–40]. Studies included in our systematic review are summarized in Table 2. The search flow diagram is shown in Figure 1. A total of sixteen patients reported multiple BAs due to odontogenic infection, and were included and analyzed in this review [6,26–40].

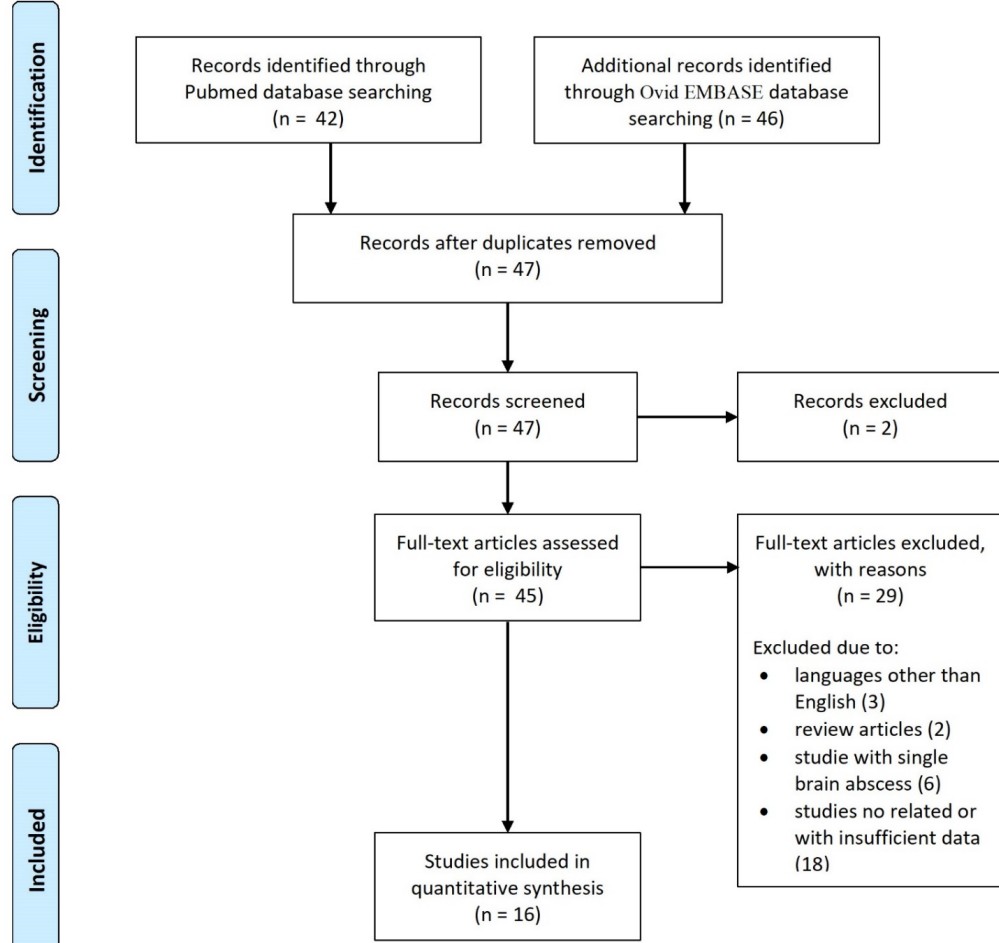

**Figure 1.** PRISMA flow diagram.

**Table 2.** Summary of studies included in the review.

| Authors | Year | Age/Sex | Neurological Clinical Presentation | N° of BAs | BAs Location | Pathogens | Surgical Treatment | Medical Treatment | Follow-Up (Months) | Outcome |
|---|---|---|---|---|---|---|---|---|---|---|
| Marks et al. [33] | 1988 | 26/M | confusion and reduced consciousness | 5 | Right frontal (3), right parietal (2) | Streptococcus viridans | - | - | - | Death |
| Kuijper et al. [32] | 1992 | 44/M | right arm paresis | 2 | Left frontoparietal region, right occipital | Actinomyces meyeri, Aggregatibacter actinomycetem-comitans | Drainage of the lesions | Amoxicillin | 12 | Good |
| Chacko & Chandy [27] | 1997 | 60/M | drowsy, confusion | 6 | Right frontal (1), left and right parietal lobes (5) | - | Drainage of the lesions | Penicillin, chloramphenicol, metronidazole, ampicillin, trimethoprim-sulfamethoxazole | 4 | Good |
| Stepanović et al. [38] | 2005 | 47/M | headache, nausea, vomiting, progressive left hemiparesis | 3 | Right occipital (3) | Aggregatibacter actinomycetem-comitans | Drainage of the lesions | Ceftriaxone, amikacin, metronidazole | 1 | Good |
| Ewald et al. [29] | 2006 | 49/M | right arm paresis, left leg paresis, seizures | more than 2 | - | Fusobacterium nucleatum | Drainage of the lesions | Clindamycin, metronidazole, cefuroxim | 12 | Good |
| Rahamat-Langendoen et al. [36] | 2011 | 42/M | confusion and reduced consciousness | 3 | Right parietal (2), left frontal (1) | Aggregatibacter actinomycetem-comitans | Drainage of the lesions | Antibiotic treatment | 12 | Good |
| Azenha et al. [26] | 2012 | 70/M | headache, dizziness, nausea, mild fever, left hemiparesis | 3 | Left frontal, right occipital, left thalamic | Streptococcus viridians, Bacteroides | Drainage of the lesions | Ceftriaxone, amoxicillin, metronidazole | 12 | Good |

**Table 2.** *Cont.*

| Authors | Year | Age/Sex | Neurological Clinical Presentation | N° of BAs | BAs Location | Pathogens | Surgical Treatment | Medical Treatment | Follow-Up (Months) | Outcome |
|---|---|---|---|---|---|---|---|---|---|---|
| Clifton et al. [28] | 2012 | 56/M | nausea, vomiting | 2 | Right frontal, left frontal | - | Drainage of the lesions and dental extraction | Vancomycin, aciclovir, ceftriaxone | 3 | Good |
| Wu et al. [40] | 2014 | 32/M | left arm paresis, left facial palsy | 3 | Right frontal, right temporal, right basal ganglia | Prevotella denticola | Drainage of the lesions | Cefepime, penicillin, metronidazole | 6 | Good |
| Pallesen et al. [35] | 2014 | 55/M | left leg paresis | 3 | Right frontal, right occipital, left occipital | Streptococcus intermedius, Staphylococcus warneri | Drainage of the lesions | Ceftriaxone, vancomycin | 6 | Left leg paresis |
| Martiny et al. [34] | 2017 | 66/M | dysarthria, diplopia, nystagmus, right peripheral facial palsy, right deafness | 3 | Right cerebellopontine angle, cerebellum | Campylobacter rectus | Drainage of the lesions and dental extraction | Meropenem, doxycycline | 10 | Good |
| AlHarmi et al. [6] | 2018 | 8/F | fever, left hemiparesis, hyporeflexia | 5 | Right frontal (3), left parietal (2) | - | Drainage of the lesions | Meropenem, vancomycin | 1 | Left hemi-paresis |
| Viviano & Cocca [39] | 2018 | 28/M | headache, nuchal pain, vomiting, confusion | 2 | Left parietal, left occipital | Streptococcus intermedius, Actinomyces | Drainage of the lesions | Vancomycin, ceftriaxone, metronidazole, clindamycin, ampicillin | 6 | Good |
| Ryan et al. [37] | 2019 | 79/M | progressive right arm paresis | 2 | Left parietal, right frontal | Blastomyces dermatitidis | Drainage of the lesions and dental extraction | Vancomycin, ceftriaxone, metronidazole | 6 | Good |

**Table 2.** *Cont.*

| Authors | Year | Age/Sex | Neurological Clinical Presentation | N° of BAs | BAs Location | Pathogens | Surgical Treatment | Medical Treatment | Follow-Up (Months) | Outcome |
|---|---|---|---|---|---|---|---|---|---|---|
| Jung et al. [31] | 2019 | 45/M | right facial spasms, tingling of the right arm, paresthesia, dysarthria | 3 | Right frontal (2), left frontal (1) | Streptococcus anginosus | Drainage of the lesions and dental extraction | Metronidazole, cefotaxime, ceftriaxone, Vancomycin, Augmentin | 3 | Good |
| Gemelli et al. [30] | 2020 | 71/F | confusion, disorientation in time and space with inattention | more than 10 | Frontal and parietal lobes | Aggregatibacter aphrophilus | Dental extractions | Vancomycin, piperacillin/tazobactam, ceftriaxone | 2 | Good |

BAs, brain abscesses; F, female; M, male.

### 3.2. Demographic, Clinical and Radiological Characteristics

Overall, the median age of patients was 48.6 years (range 8–79) and the proportion of male patients was 87.5%. Only two patients were female. In most of the cases (75%) multiple BAs were the result of periodontal disease or multiple dental caries. Multiple BAs followed professional tooth cleaning in two cases [35,39] and dental extraction in one case [40]. The most common clinical neurological presentations were the onset of a motor deficit (62.5%) and confusion and reduced consciousness (37.5%). Headache and nausea were present in 18.8% of cases. One patient reported seizures [29]. The median overall number of BAs was 3.7 and the most common locations were the frontal (46.7%), parietal (30%), and occipital (13.3%) lobes, whereas cerebellum (5%), basal ganglia (3.3%) and temporal lobe (1.7%) were rare locations. The right hemisphere was more involved (63.3%) than the left hemisphere (36.7%). In all cases, BAs appeared as well-defined, oval-shaped and hypoechoic lesions, which were isointense to the muscle on T1 images and hyperintense on T2 images on magnetic resonance imaging (MRI). Post-contrast lesions showed homogeneous enhancement in all of cases.

### 3.3. Pathogens, Treatment and Clinical Outcome

The most common pathogens involved in the development of multiple BAs were Streptococcus viridans (with its different species), that was identified in 31.3% of cases, and Aggregatibacter actinomycetemcomitans (18.8% of cases). Other pathogens identified in single cases were Campylobacter rectus, Fusobacterium nucleatum, Prevotella denticola, Staphylococcus warneri, Actinomyces meyeri and Blastomyces dermatitidis, present in 50% of Gram-positive bacteria and 50% of Gram-negative bacteria.

A total of 14 out of 16 patients (87.5%) underwent craniotomy and surgical drainage of at least one of the multiple BAs in concomitance with antibiotic therapy, whereas five of these patients also underwent dental extractions (31.3%). One patient [30] underwent dental extractions and antibiotic therapy without surgical drainage of BAs and one patient [33] died without any surgical cranial or oral treatment.

The most common antibiotics used were cephalosporins (66.7% of cases), metronidazole (53.3%), vancomycin (46.7%) and penicillin (46.7%). Median follow-up was 6 months. A total of 13 patients (81.3%) had a good outcome, whereas two patients (12.5%) showed persistent deficits in motor skills (paresis) at last follow-up. One patient died during hospitalization; therefore, overall mortality and morbidity of multiple BAs was 6.3% and 12.5%, respectively.

## 4. Discussion

### 4.1. History and Epidemiology

Sir Percival Pott was probably the first to recognize and document that infections elsewhere in the body could spread and cause a BA [41,42]. Then, the French surgeon Morand, in 1768, made the first report of successful surgical treatment of an otitic BA with good recovery [43]. In 1891, Topuzlu treated surgically, for the first time, a BA that originated as a complication of a depression fracture of the cranial inner table, using contemporary anesthesiological and surgical techniques [44,45]. It has been postulated that oral microorganisms may enter the cranium by direct extension (hematogenous spread, local lymphatics) or indirectly by extraoral odontogenic infection [46]. Hematogenous spreading can occur through general circulation or along the facial, angular, ophthalmic or other veins which lack valves, through the cavernous sinus and into the cranium [35,47]. Pallesen et al. [35] described both modes of infection dissemination, either by means of direct extension via fascial planes or by means of hematogenic or lymphatic spread. Immunodeficiency due to HIV infection, alcohol abuse, diabetes, chemotherapy or cancer may promote the development of BAs [48–50].

Oral infections are a rare cause of BAs, contrary to general belief and, in most cases, dental radiography does not help to make a proper diagnosis; it is difficult to accurately diagnose odontogenic disease due to its occurrence as an acute inflammatory progression

of the soft tissue around the alveolar bone [31]. In support of a medical diagnosis, recent papers reported the role of prognostic early markers in the diagnosis of oral diseases [21–23]. Patients with periodontitis presented significantly higher serum and salivary Galectin-3 levels in comparison to healthy subjects, suggesting that periodontitis and Endothelin-1 were the significant predictors of serum and salivary Galectin-3 levels, respectively [21]. With a suspected BA, further diagnostic imaging studies, such as, at first, a head computed tomography (CT) scan, should be taken into account [51–53]. In BAs with a suspected remote infection, it is reasonable to carefully evaluate the head and neck areas close to the brain and chronic infectious diseases such as odontogenic infection that can cause such pathology [1,31,54]. A total of 40% of BAs are caused by chronic otitis media or mastoiditis, 10% by maxillary sinusitis or paranasal sinusitis, and 50% through the spread of cardiac or pulmonary infections [31]. In a few cases (5–7%), BAs can have an odontogenic origin that can occur after dental extraction, but also after professional tooth cleaning, mostly in patients with poor oral health, as we reported. Bacteremia's occurring even as a result of routine daily dental interventions such as toothbrushing has been well documented, although its significance remains unclear [55,56]. Although diet and the environment have a great impact on gut microbiota [57], they exert minimal effect on the composition of oral bacteria and oral microbiota. However, because of ecological interactions and environmental conditions, over time, the oral anaerobes eventually become the predominant group in the endodontic and periapical infections [58,59]. Anaerobic infections (bacilli and Gram-positive cocci) are associated with incidence of acute signs and symptoms, such as pain, sensitivity to pressure and cellulitis and are characterized by abscess formation, foul-smelling pus and tissue destruction [58]. Some previous studies highlighted the role of the oral microbiota in the colonization of dental implants [60,61]. Through bacterial cultures, it has been seen that the quality of microorganisms present in the oral cavity before the surgical implants determines what will subsequently populate the implant itself. Furthermore, it has been shown that the biofilm present in peri-implantitis has a composition similar to that of periodontitis, with high levels of periodontal pathogens, increasing the risk of peri-implantitis in patients with a past history of periodontitis and increasing the potential risk of hematogenous dissemination and BAs [60,62].

### 4.2. Pathogen Features and Clinical Presentation

Most dental abscesses are caused by the resident oral microbiota that enters normally sterile tissues. The major isolates are streptococci and anaerobic bacteria, which are regarded as normal flora of the tooth and gingival crevice [63]. The microbiota specificity in odontogenic infections has been more clearly delineated with technological advances in sampling and anaerobic culture. Li et al. [46] reported that the most frequent species isolated in BAs are Gram-positive cocci (Streptococcus mutans, Streptococcus milha, Streptococcus intennedius) and Gram-negative rods (Aggregatibacter actinomycetemcomitans, Prevotella oralis and Fusobacterium nucleatum). Laulajainen-Hongisto [64] reported that the most common pathogens involved in intracranial BAs were Streptococcus (42%), Fusobacteriae (14%), Aggregatibacter actinomycetemcomitans (8%) and Staphylococcus spp (8%). Similarly, we reported that the most common pathogens involved in the development of multiple BAs were Streptococcus viridans (with its different species), which was identified in 31.3% of cases, and Aggregatibacter actinomycetemcomitans (18.8% of cases), although Gram-positive and Gram-negative bacteria seem to be involved in equal part, as we found in this review. Anaerobic infection is favored in the brain due to the low oxygen tension of the interstitium and because the BA causes focal infarcts due to decreased oxygen supply [46]. Streptococcus intermedius, which is a member of the Streptococcus anginosus group, despite being a part of the normal microbiota, is one of the most common pathogens associated with brain and liver abscesses and thoracic empyema, increasing, as a result, the morbidity and mortality rates in affected patients [65]. Some oral bacteria such as Aggregatibacter actinomycetemcomitans is non-motile, facultative anaerobic, small Gram-negative coccobacillus that shows a predilection for the central nervous system when

they produce systemic disease [46,66]. However, it is difficult to isolate through culture due to its fastidious slow-growing nature, especially because it is often part of mixed infection with other bacteria [36,67,68]. As our review confirmed, BAs were often caused by poly-microorganisms (in 25% of cases) and for this reason it is difficult to identify accurate pathogens [31].

In the case of a single BA, the presenting features and clinical neurological presentations depend on the size and intracranial location of BA, the virulence of the infecting agents, the immunologic status of the host, and the cerebral edema caused by the expanding intracranial mass lesion [28]. The classic triad of fever, headache and focal neurologic deficit is present in less than 50% of cases [28]. In the case of multiple BAs, the classic triad of fever, headache and focal neurologic deficit is present in more than 63% of cases. Similarly, mental status change and reduced consciousness is present in up to 37.5% of patients, as we reported, often accompanied by the acute onset of meningeal signs, suggesting either herniation or intraventricular rupture of the abscess.

### 4.3. Differential Diagnosis and Outcome

In addition to a clinical examination, diagnostic imaging should be used to diagnose BAs. The diagnosis of BAs is sustained by contrast-enhanced MRI, where the abscesses appear as typical ring-like structures surrounded by edema. Differential diagnosis between BA and tumor is of extreme importance, infection being differentiated from neoplastic disease by virtue of hyperintense signals in diffusion-weighted MRI scans [35]. Management of multiple BAs typically involves surgical drainage of at least one of the BAs and simultaneous eradication of the primary odontogenic source of sepsis in association with intravenous administration of high doses of antibiotic agents [28,31,34,37].

Antibiotics are often changed (100% of cases) during the entire hospitalization for multiple BAs in order to find the most appropriate treatment. When antimicrobial sensitivity patterns are determined, treatment is then tailored based on the result. In patients with multiple small BAs who are neurologically intact, the disease is managed medically, with close serial monitoring performed using neuroimaging to determine the response to antibiotic therapy [33]. Otherwise, patients with neurological deficits may require urgent evacuation of the lesion [36,69]. For single BA, the proportion of patients with good outcomes enabling return to prior occupation rose over time, from 12% in 1970–1989 to 24% in 1990–2012 [48]. Patients with multiple BAs experienced a good recovery (81.3%) with a morbidity rate of 12.5%.

There is no definite rule for the treatment of BA. Antibiotics are considered to be the first line of treatment and should be started immediately. It has been reported that the Streptococcus anginosus group is sensitive to penicillin, ampicillin, cephalosporin, clindamycin and erythromycin, and that Aaggregatibacter actinomycetemcomitans [31] is sensitive to strong synergistic effects with metronidazole in combination with amoxicilline [70]. Medical treatment alone is not recommended for pyogenic BAs greater than 2.5–3 cm in diameter and for refractory smaller BAs [6,58]. Surgical options include open craniotomy with complete excision and stereotactic aspiration. Open surgery is indicated for large and multiple lesions, cerebellar lesions which obstruct CSF flow and fungal abscesses, and contraindicated in deep-seated lesions [6]. Aspiration is associated with reduced morbidity; however, up to 70% of patients require a repeated procedure due to factors such as incomplete aspiration and inadequate antibiotic coverage [6,58]. When multiple abscesses are present, a successful strategy dictates aspiration of lesions >2.5 cm and treating other lesions conservatively for 6–8 weeks [6].

Oral pathogens are necessary, but not sufficient for odontogenic BAs. The role of microbiota is strictly connected with the immune system that appears to be the crucial determinant of disease susceptibility and severity [46,58]. Cardiovascular disease, chronic obstructive respiratory disease, rheumatic disease, diabetes, chronic inflammatory bowel diseases, cancer and organ transplants are factors that affect patients rendering medically immunocompromised and the oral microbiota [46,71–76]. BA can result in permanent

neurological sequelae such as epilepsy, cranial nerve palsies, intellectual and behavioral disorders, motor deficits, hydrocephalus and death. Early diagnosis and intervention can lead to reduced mortality rates and improved outcomes [6,77]. Poor prognosis is reported in immunocompromised patients, children younger than 1 year of age, in delayed diagnosis, in the case of coma or severe mental status change at the time of diagnosis and when BAs rupture into ventricles [6,78–82]. Prevention of abscesses by routine dental and oral hygiene care or early removal of abscessed or non-treatable teeth is prudent and appears to be more important in the immunocompromised patient.

Our review has several limitations. The reported cases are retrospective and only single case reports. Second, our review might be underpowered because this topic is seldom reported in the literature and subjected to continuous updating. Lastly, the role of oral microbiota in the onset of multiple BAs is still under investigation. For this reason, larger studies are required to analyze the relationship between multiple BAs and oral microbiota.

## 5. Conclusions

Dental infection is an uncommon source of multiple BAs, often but not always seen in immunocompromised patients. Streptococcus viridans and Aggregatibacter actino-mycetemcomitans seem to be the most common pathogens involved in the development of multiple BAs, but at the same time bacteremia (depending upon several factors such as local oral inflammation, oral microbiota and systemic disease) play an important role in their development. A detailed history taking and physical examination, as well as early detection of the lesion with prompt medical and surgical treatments, can lead to an early diagnosis and a successful management of multiple BAs, reducing the duration of empirical antibiotic therapy and the time of hospitalization. Serum and salivary Galectin-3 levels were demonstrated to be a valuable prognostic early marker of periodontitis. A multidisciplinary approach is essential in the management of multiple BAs. Further studies are required to better understand the role of microbiota in the development of multiple BAs.

**Author Contributions:** Conceptualization, N.M., B.R. and P.P.; methodology, N.M. and B.R.; data collection, N.M. and B.R.; formal analysis, N.M. and B.R.; writing—original draft preparation, N.M., P.P. and B.R.; investigation, N.M., B.R. and B.C.; resources, B.R.; data curation, N.M, B.R. and W.M.; writing—review and editing, N.M., B.R., P.P., M.C. and A.S. All authors have read and agreed to the published version of the manuscript.

**Funding:** This research received no external funding.

**Institutional Review Board Statement:** Not applicable.

**Informed Consent Statement:** Not applicable.

**Data Availability Statement:** Data sharing not applicable.

**Acknowledgments:** We thank Federica Tataranni for her English revision.

**Conflicts of Interest:** The authors declare no conflict of interest.

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
