# Peer review of "Multiple Brain Abscesses of Odontogenic Origin. May Oral Microbiota Affect Their Development? A Review of the Current Literature"

_applsci, doi:10.3390/app11083316_

Round 1
Reviewer 1 Report
Abstract and Introduction
The title precisely introduces to the subject of the article. The aims of the paper are clearly explained and the content of the paper remains in line with the objectives. The abstract is concise, giving a proper description of a subject. The originality of the topic ought to be appreciated, because no other authors had presented before a thorough systematic review regarding the oral microbiota as a etiological cause of intracranial complications.
Material and methods
The systematic review was preformed on the basis of PRISMA guidelines, thus it provides a valuable source of information about multiple brain abscesses with odontogenic origin. Authors have analyzed Pubmed and Ovid EMBASE using the proper inclusion and exclusion criteria. 16 chosen papers are relevant to the subject. The methodology is explicit and its presentation with the PRISMA flow diagram helps the reader to be better oriented in the research area.
Results and discussion
The summary of the review is presented in a properly designed table. What is important, indeed it reveals the spectrum of oral microorganisms that were the etiological factor of multiple BA.
It needs to be emphasized that topic is relevant according to high mortality of BA. The life-threatening or fatal complications of this condition can be avoided when the early diagnosis and treatment is implemented.
The paper describes thoroughly not only the epidemiology, but also the bacterial etiology of BA with a particular attention to the intraoral pathogens and their role in intracranial complications. Full spectrum of possible symptoms is presented. It can increase the awareness among not only general practitioners or neurologists, but also among dentists, because these specialists are the first to restrain odontogenic infection. I appreciate the part of discussion about differential diagnosis and outcome, because it gives the current therapeutic guidelines including recommended antibiotic therapy and surgical treatment. Unfortunately there is a lack of information about recommended doses and routes of drugs administration. However it can be explained, because authors underlined the need of tailored-made therapy
The authors are aware of the limitations of their research that is underlined in the final paragraph of discussion part.
The reference list is quantitatively impressive (77 positions) The Conclusions section is brief and refers to the aim of the study. The language is fluent and precise.
To sum up suggest publication after the authors have considered the following mostly minor remarks
Line 23, 39 morbidity – what does it mean “high morbidity” - 0.3 to 1.3 cases per 100.000?
Line 37 should be “is converted”
Line 48 – sentence should be rewritten
Line 51 – is speak a proper word? Should be: report./inform etc.
Line 58 unnecessary ,
197 isolated
Some long sentences require minor editing and should be divided into shorter part i.e line 223-228,, 237-242, 246-249.
Author Response
Thank you for your comments.
- Lines 23-39. We changed “high morbidity” with “long term morbidity”, meaning impairment, or degradation of health that affect quality of life in long term in those patients experience brain abscess.
- Lines 37, 48, 51, 58, 197 - As you recommended, we revised the text.
- Lines 223-228, 237-242, 246-249 - As you recommended, we divided those sentences in more parts.
Reviewer 2 Report
I enjoyed reading your manuscript. There is merit to it.
The written style and presentation may be improved. This manuscript has been well conducted and reported. I believe it will have a better reach with the minor corrections that have been suggested.
This is well-written manuscript. Although we can take from the conclusions for clinical significance, we have the moral obligation to report the current status of such a topic.
Author Response
Thank you for your comments. As you recommended, we revised and improved the manuscript according with reviewer’ suggestions.